# Veterinary Professionals’ Understanding of Common Feline Behavioural Problems and the Availability of “Cat Friendly” Practices in Ireland

**DOI:** 10.3390/ani9121112

**Published:** 2019-12-10

**Authors:** Matt Goins, Sandra Nicholson, Alison Hanlon

**Affiliations:** School of Veterinary Medicine, University College Dublin, D04W6F6 Dublin, Ireland; lewis.goins@ucdconnect.ie (M.G.); sandra.nicholson@ucd.ie (S.N.)

**Keywords:** veterinary behavioural medicine, feline behaviour, veterinary behaviour curriculum, feline welfare, behaviour, aggression

## Abstract

**Simple Summary:**

Veterinary behavioural medicine, which includes being able to understand animal behaviour and treat behaviour problems, is an important part of veterinary practice. However, many veterinary practitioners and veterinary nurses in Ireland and elsewhere feel that they have received inadequate training in this subject. The purpose of this study was to survey veterinary practitioners and veterinary nurses in Ireland about treating common behavioural problems in cats and the availability of “cat friendly” practices. An online survey was developed, consisting of 21 questions on professional roles and experience, scenarios presenting advice given on common cat behaviour problems, and “cat friendly” practice management options. For each piece of advice participants were asked to score how likely it would be to solve the behavioural problem in a kind way. The online survey was shared via professional organisations, social media and at the University College Dublin Hospital Conference. The survey was completed by 42 veterinary practitioners and 53 veterinary nurses. Most of these correctly recognised both good and bad advice, but some mistakes and uncertainties were found. The scores of veterinary practitioners and veterinary nurses differed for the advice on urine spraying, self-mutilation (self-injury), and resource-based aggression (aggression related to sharing items), and we found that relatively few “cat friendly” measures were available in respondents’ clinics. Our findings could be used to improve training in veterinary behavioural medicine.

**Abstract:**

Veterinary behavioural medicine (VBM) is an integral aspect of veterinary practice. However, Golden and Hanlon (Ir. Vet. J. 71:12, 2018) found that the majority of professionals surveyed felt they had received inadequate VBM education and were commonly asked to give advice on feline behavioural problems. The purpose of this study was to explore understanding of feline VBM and the availability of “cat friendly” provisions in clinical practice in Ireland. An online survey comprised 21 questions on professional role and experience, vignettes of common feline behavioural problems, and “cat friendly” practice management. Using a Likert Scale, participants were requested to score whether the advice depicted in vignettes supported best outcome based on the definition by Shalvey et al. (Ir. Vet. J. 72:1, 2019). The survey was distributed via professional organisations, social media, and at the University College Dublin Hospital Conference. Forty-two veterinary practitioners (VPs) and 53 veterinary nurses (VNs) completed the survey. The majority of veterinary professionals agreed with our classification of best outcome, but some areas of disagreement and uncertainty were identified. In addition, there were significant differences between VPs and VNs regarding spraying (*p* = 0.033), self-mutilation (*p* = 0.016), and resource-based aggression (*p* = 0.013). Relatively few “cat friendly” measures were implemented in respondents’ clinics. Our findings support the need for increased education in feline VBM, in particular, implementation of cat friendly practice initiatives.

## 1. Introduction

Veterinary behavioural medicine (VBM) is an integral aspect of veterinary practice; in particular, for companion animal welfare. However, Golden and Hanlon [1] found that the majority of the veterinary professionals they surveyed in Ireland considered their training to be inadequate in VBM. This was a view shared by respondents in a study by Kogen et al. [2] in which only 42.8% reported “a significant amount” of veterinary behaviour training as veterinary students while 39.4% and 17.8% reported receiving “a few hours” and no training respectively, despite the fact that the majority (99.4%) had seen canines or felines with behavioural issues while working in practice. This lack of education and training extends to other disciplines in veterinary medicine. Mota-Rojas et al. [3] found that less than half of the veterinary programmes surveyed throughout Latin America had any compulsory education in animal welfare and behaviour. Shivley et al. [4] found similar results for the American Veterinary Medical Association (AVMA) accredited institutions, with only three-quarters of the US programmes having a behaviour course on offer, which was compulsory in nearly two-thirds of those programmes.

Sandoe et al. [5] describe a transformation in the companion animal practice before which felines were treated as small dogs rather than as a species in their own right [6]. Veterinary practice continues to evolve with the emergence of cat-only clinics corresponding to an increase in cat ownership and a greater demand for services to support this demographic. Golden and Hanlon [1] found that veterinary professionals in Ireland routinely received queries regarding feline behaviour. Additionally, the People’s Dispensary for Sick Animals [7] reported that a majority of cat owners (77%) surveyed in the UK stated that their cat displayed a behaviour that they would like to change and 89% of cats expressed a fear-related behaviour. Owners were concerned about inappropriate scratching (49%), being woken up (17%), begging for food (17%), aggression (9%), and inappropriate toileting (8%) [7]. This is important because unwanted or antisocial feline behaviour can be due to illness, stress, or lack of socialisation, any or all of which can impact on their welfare needs. However, clients most commonly seek veterinary advice for feline house soiling and inappropriate scratching [8].

Whilst Golden and Hanlon [1] demonstrate a client interest in feline behaviour, there are relatively few “cat friendly clinics” registered in Ireland, which may reflect a divergence between client demand and clinical provision of feline VBM. The aim of this project was to explore veterinary professionals’ understanding of advice to support the best outcome for common feline behavioural problems where the best outcome was defined as one which provided a resolution to the behavioural problem while not compromising the animal’s welfare [8]. An additional project aim was to capture the prevalence of “cat friendly” initiatives in veterinary practice in Ireland. This survey serves as the third part of a broader research aim to develop day-one competencies for VBM and to inform training in veterinary medicine and veterinary nursing in Ireland [1,9].

## 2. Methods

### 2.1. Survey Design

An online survey was designed to explore veterinary professionals’ understanding of advice to support best outcome for common feline behavioural problems and to capture the prevalence of “cat friendly” initiatives in veterinary practices in Ireland. It consisted of 21 questions, divided into three sections: professional role and experience, scenarios (vignettes) of common feline behavioural problems, and “cat friendly” practice management.

Section one consisted of consent to participate in the study and questions on professional role to understand the respondent demographic (such as the profession of the respondent (veterinary practitioner (VP), veterinary nurse (VN), or other). Those who selected “Other” were permitted to complete the survey; however, their data was left out of the analysis. Year of graduation, origin of veterinary training, and confidence in addressing feline behaviour problems were also requested in this section.

Section two presented 10 vignettes, which depicted advice from a veterinary professional (VP or VN) regarding a common feline behaviour. The behavioural problems used in the vignettes were selected by drawing upon recent research by Golden and Hanlon [1] as well as current literature [7,10,11,12] and the authors’ experience. Vignette development is described in Section 2.2. They were designed to illustrate common scenarios that were either likely (questions 8, 9, 11, 12, and 16) or unlikely (questions 7, 10, 13, 14, and 15) to result in the best outcome for the cat(s) in question. Respondents were asked whether the advice offered by the veterinary professional in each vignette would support the best outcome using a Likert scale (“Extremely Likely”, “Likely”, “Neither Likely nor Unlikely”, “Unlikely”, “Extremely Unlikely”, or “Don’t Know”). The best outcome was defined as that which provided a resolution to the behavioural problem while not compromising the animal’s welfare [8]. A comment box was provided with each question in this section to enable respondents to qualify their response.

The final section of the survey focused on “cat friendly” clinical practice, aspects of the facilities and protocol for the waiting area, consultation room, hospitalisation, and boarding that are likely to either cause stress, fear, and/or anxiety in cats or alleviate these responses based upon recent literature [9,13] and initiatives such as Fear Free Pets^®^. In relation to the reception area, respondents were asked if their clinic offered any of the following: a cat-only reception/entrance/waiting area, cat-only consultation hours, shelves for cat carriers above “dog level,” towels/covers for cat carriers, and televisions, magazines, or other materials in waiting areas for owners. In relation to the cat ward, respondents were asked about design and management practices available in their practice that may either increase or decrease the stress and anxiety of patients. The options that could potentially improve the wellbeing of patients were a separate cat ward and a set routine each day for the cat ward. Whereas those that could potentially increase stress and anxiety included having machinery located in the cat ward, needing to carry cats through the dog ward to access treatment areas, patients having visual contact with unfamiliar conspecifics, and dogs being walked through the cat ward for treatments and toileting. The final questions in this section asked respondents if their clinic boarded cats and whether boarded cats were kept in a separate space to patients. A comment box was provided with each question in this section to enable respondents to qualify their response.

The survey was sent out for peer review to two veterinary practitioners (O.D. and S.N., both of whom have postgraduate qualifications in animal behaviour) and E.S., first author of Shalvey et al. [8], a sister study on canine VBM. Based on their feedback, amendments were made prior to distribution of the survey via SurveyMonkey^®^.

### 2.2. Vignette Development

The vignette methodology was adapted from several studies, including Collins et al. [14], in which it was used to explore stakeholder perception of equine welfare; Magalhaes-Sant’Ana and Hanlon [15], which used it to support student learning in veterinary ethics; and Shalvey et al. [8], in which it was used to investigate veterinary professionals’ understanding of common canine behavioural problems. In the current study, vignettes were used to depict the advice provided by a veterinary professional (either a VP or VN) in response to a client query regarding a common feline behavioural issue.

Feline behavioural issues were identified from the scientific literature, a report published by the People’s Dispensary for Sick Animals [7], and the authors’ experience. The behaviours selected were: house soiling, destructive behaviour (e.g., scratching furniture, carpets), play-related aggression towards owner or other family members, aggression towards other pets, self-mutilation (overgrooming), anxiety-related issues, and fear-related issues for the vignettes.

A review of the scientific literature was conducted to determine the current best practice approaches (and less effective approaches) to the treatment of each behavioural problem. We adopted the same methodology as Shalvey et al. [8] where survey participants were asked to score, ‘How likely is this recommendation to give the best outcome?’ Shalvey et al. [8] defined “best outcome” as “one which provided a resolution to the behavioural problem while not compromising the animal’s welfare.” Five vignettes were designed to support best outcome and the remaining five were unlikely to achieve this goal (Table 1).

Vignettes 2 (spraying), 3 (destructive behaviour), 5 (anxiety—child related), 6 (anxiety—moving house), and 10 (aggression—cat/cat resource-based aggression) depicted the best outcome scenarios based on the following rationale. Vignette 2 (spraying) was considered likely to support best outcome because it incorporated a multifaceted approach to limit spraying, including neutering [11], use of a non-ammonia based cleaner to remove olfactory signals, and application of pheromones to replace spray marks, and a physical barrier to impede the neighbourhood cat from coming into the garden [11]. The advice to address the scratching of furniture depicted in Vignette 3 was considered likely to support best outcome because it aimed to redirect this normal but destructive behaviour in a positive way using a combination of pheromone treatments and catnip [13,16]. Vignette 5 (anxiety—child-related) was considered likely to support best outcome as the suggested measures would reduce stress/fear by facilitating the cat’s avoidance behaviour and giving it a sense of control over its environment. Feeding the cat within the territory would also enhance the desirability of the “escape” area and foster positive emotions [17]. However, in the long term, behavioural counselling would be needed to modify this behaviour. The advice in Vignette 6 (anxiety—moving home) was considered likely to support best outcome because it attempted to reduce the anxiety associated with a change in physical environment by use of the same pheromone treatments (plug-ins) in the current and new house; thus, providing the cat with familiar anxiety-reducing olfactory cues in the new house [13]. Provision of sufficient resources in multicat households was the focus of Vignette 10 (aggression—cat/cat resource-based aggression). The advice was considered likely to result in best outcome because it considered both increasing the number of resources and directing the location of the new resources into the specific territory of the bullied cat in the vignette [13,17]. However, as always, the general rule of thumb when providing resources should be “one for each cat plus one” [13].

The remaining five vignettes (1, 4, 7, 8, and 9) were designed to illustrate advice that was unlikely to result in best outcome based on the following rationale. Vignette 1 (inappropriate toileting) was considered unlikely to support best outcome because although the new litter box location would provide privacy and seclusion [18], the use of an ammonia based cleaner may encourage the cat to continue to toilet in the inappropriate area, and thus, the advice is unlikely to resolve the problem [9,18]. Self-mutilation, as described in Vignette 4, is a complex issue. The recommended treatment would help to prevent further injury, but it would not address the underlying cause. As a result, the self-mutilation would be likely to persist or to recur once the bandages are removed [19]. The advice given to the owner in Vignette 7 (fear—loud noises) was unlikely to support the best outcome because reassurance may be ineffective or may reward the behavioural manifestations of fear. In the short term, the cat should be allowed to choose other coping mechanisms, such as retreat and hiding [9,13]. Anxiolytic medications or pheromones may also be helpful for calming the cat. In the medium to long term, desensitisation and counterconditioning would be most effective [9]. Vignette 8 (fear—strangers) was considered unlikely to result in best outcome because insufficient details were provided on how to perform kitten socialisation. During socialisation, kittens should be introduced to wide range of different individuals. However, interaction should be voluntary and modified or ceased if the kitten displays fearful behaviour. Otherwise, flooding could occur, resulting in stress and a greater fear of strangers [8]. Finally, Vignette 9 (aggression—play related) describes a positive punishment technique used to thwart normal play behaviour in a kitten. Use of aversion learning may result in unintended and undesirable behavioural outcomes, such as avoidance of the owner. To support best outcome the owner could be advised to redirect the behaviour using toys [13].

### 2.3. Ethical Approval and Administration of Survey

The nature of the survey qualified it for an exemption from full ethical approval by the Human Research Ethics Committee at University College Dublin (LS-E-19-79-Goins-Hanlon). A copy of the survey can be found in the Appendix A.

The survey was published online using SurveyMonkey^®^ and was open for responses from the 1 July 2019 until the 22 July 2019. An invitation to complete the online survey was distributed by Veterinary Ireland, the Irish Veterinary Nurses Association (IVNA), and by XL Vets, Ireland. In addition, the survey was shared on social media via University College Dublin School of Veterinary Medicine Twitter (@ucdvetmed) and @AlisonHanlon and the University College Dublin Veterinary Hospital Facebook page (https://www.facebook.com/ucdvet/). The survey was also publicised at the University College Dublin Veterinary Hospital Conference on 11 July 2019.

### 2.4. Statistical Analyses

Data were exported from SurveyMonkey^®^ into Microsoft Excel (2013). R version 3.6.1 and R Studio version 1.2.1335 [20] were used for data cleaning and transformation, data visualisation, generating descriptive statistics, and for all statistical analyses. Microsoft Excel (2013) was also used to generate graphs. To test for statistically significant differences in the distributions of responses between independent cohorts, a Wilcoxon rank sum test was conducted. “Don’t Know” data were excluded from these analyses, as the Wilcoxon rank sum test requires ordinal data. In addition, extremely likely and likely and extremely unlikely and unlikely responses were pooled, providing three categories for analyses: (1) likely, (2) neither likely nor unlikely, and (3) unlikely. Comparisons were made between the responses of VPs and VNs, those educated in Ireland and outside of Ireland, respondents who were very confident (>75%) and not confident (<50%) in addressing feline behavioural issues, and those who worked in clinics incorporating “cat friendly practices” and those who did not. The significance threshold for statistical analyses was *p* < 0.05.

## 3. Results

### 3.1. Respondent Demographics

A total of 147 individuals accessed the survey and 97 (66.0%) individuals completed all three sections. Respondents comprised 42 (43.3%) VPs and 53 (54.6%) VNs. Three participants selected “Other” and were excluded from the analysis. On average, it took ten minutes to complete the survey. The majority of VNs graduated after 2011, while more VPs graduated in the 1990s than any other decade (Figure 1).

Three VNs and eight VPs were trained outside of Ireland (Australia, Germany, Hungary, Italy, UK, and USA). A Wilcoxon rank sum test was performed to compare the responses of those who received their veterinary education either in the Republic of Ireland or elsewhere. No significant differences were found.

### 3.2. Confidence with Addressing Cats’ Behavioural Problems

Each respondent was asked to judge their confidence (100 point scale) in advising clients about feline behavioural issues. Confidence levels ranged from 10% to 93% for VPs and 20% to 100% for VNs. The mean confidence levels for VPs and VNs were 61.5% and 63.4%, respectively. Year of graduation had no significant effect on level of confidence of veterinary professionals (Figure 2).

A Wilcoxon rank sum test was performed to compare the responses of veterinary professionals who were either confident (>75) or less confident (<50) with addressing feline behavioural problems. No significant differences were found between these cohorts across all vignettes.

### 3.3. Likelihood of Correctly Categorising Vignettes

The percentages of respondents correctly categorising the likelihoods of achieving the best outcomes are shown in Figure 3. At least 50% of both VPs and VNs correctly categorised each vignette. The lowest consensus on the likelihood of achieving best outcome was vignette 9 (50%; aggression—play related) for VPs, and was vignette 1 (50.9%, inappropriate toileting) for VNs. The vignettes with the highest consensuses were vignette 4 (90.5%, self-mutilation) for VPs and vignette 10 (90.6%, aggression—cat/cat resource-based aggression) for VNs.

A Wilcoxon rank sum test revealed that the percentage of VNs correctly identifying the best outcome scenarios (vignettes 2, 3, 5, 6, and 10) was significantly higher than the percentage recognising the poor-likelihood-of-best-outcome scenarios (vignettes 1, 4, 7, 8, and 9) (W = 1765.5, *p* = 0.01926). There was no significant difference for the VPs or between VPs and VNs in their ability to recognise likely or unlikely best outcomes.

In each case, the most frequently selected answer by VPs and VNs was that which aligned with the correct identification of the likely or unlikely best outcome. The vignette with the most agreement was vignette 10 (aggression—cat/cat resource-based aggression, Table 2) with a combined consensus of 81.1%. The vignette with the least agreement was vignette 9 (aggression—play related, Table 2) with a combined consensus of 52.6%.

A Wilcoxon rank sum test also revealed that the percentage of VNs responding “Neither Likely nor Unlikely” was significantly higher in those scenarios that were unlikely to support best outcome (vignettes 1, 4, 7, 8, and 9) than those that were likely to support best outcome (vignettes 2, 3, 5, 6, and 10) (W = 1196.5, *p* = 0.02555). There was no significant difference for the VPs or between VPs and VNs in the “Neither Likely nor Unlikely” response category.

The percentage of respondents who answered “Don’t Know” for each vignette ranged from 0% to 17% (Figure 4). The highest percentage of VPs that answered “Don’t Know” to a vignette was 4.8% (vignette 9, aggression—play related) while the highest percentage of VNs that answered “Don’t Know” to a vignette was 17.0% (vignette 4, self-mutilation). The greatest contrast in percentage of respondents answering “Don’t Know” was vignette 4, where 17.0% of VNs answered “Don’t Know” compared to 2.4% of VPs.

### 3.4. Comparison by Profession

A Wilcoxon rank sum test showed that there was a significant difference in the distribution of responses between VPs and VNs for vignette 2: spraying (W = 874.5, *p* < 0.05); vignette 4: self-mutilation (W = 1341, *p* < 0.05); and vignette 10: aggression: cat/cat resource-based aggression (W = 13405, *p* < 0.05). A smaller percentage of VNs (56.6%) than VPs (78.6%) correctly identified that the advice was likely to achieve the best outcome in vignette 2 (Figure 5) and the same held true for vignette 4 (Figure 6), where 69.8% of VNs correctly identified that the advice was unlikely to achieve the best outcome compared to 90.5% of VPs, while in vignette 10 (Figure 7), the majority (90.6%) of VNs correctly identified that the advice was likely to achieve the best outcome compared to 69.0% of VPs.

### 3.5. “Cat Friendly” Practices

Towels and covers for cat carriers were the only options available in the waiting areas of the majority of respondents’ clinics (52.4% VP, 60.3% VN). In relation to the consult room, 14.3% (VP) and 17% (VN) of respondent clinics provided a separate cat-only consult room, 28.6% (VP) and 39.6% (VN) used feline synthetic pheromone products during consults (such as Feliway™), and 35.7% (VP) and 39.6% (VN) had cat bags or wraps. The most common option offered during consultations was additional time to allow nervous cats to settle (52.4% VP, 28.3% VN).

While the majority of respondents (59.5% VP, 66.0% VN) reported that they had a separate cat ward, the majority (76.2% VP, 66.0%) also indicated that a set routine for the cat ward was not an option for their clinic. Potential stressors in the cat ward (listed in the survey) were mostly absent at majority of respondents’ clinics, although on average 15% of veterinary professionals indicated that dogs may be walked through the cat ward for treatment or toilet purposes and 8.4% indicated visual contact with other patients. Similarly, in relation to long term care, the majority of respondents’ clinics provided services that tend to reduce stress and anxiety in feline patients, including keeping the patient in the same cage throughout the stay (64.2% VP, 66.0% VN) and encouraging their owners to bring in bedding, food, and litter from home (47.6% VP, 62.3% VN).

Finally, the majority of respondents (52.4% VP, 56.6% VN) indicated that their clinics did not board cats, and of those who did, the majority (75% VP, 62.5% VN) boarded cats in a separate area to the feline patient ward.

The availability of cat friendly practices (e.g., shelves for cat carriers in the waiting area) in respondents’ clinics had no significant effect on responses to vignettes.

## 4. Discussion

A key challenge of the survey was gaining access to veterinary professionals in Ireland due to data protection laws. Because of this, veterinary professionals in Ireland were contacted by various professional organisations, via social media, and at the UCD annual Veterinary Hospital conference. As such, the response rate for the survey cannot be determined.

### 4.1. Veterinary Demographic

There was a difference in the demographics of VP and VN respondents, with VPs graduating mainly in the 1990s, whereas the VN cohort had mainly graduated since 2011. A similar result was recorded by Shalvey et al. [8] and can be partly explained by the relatively recent introduction of the veterinary nursing degree in Ireland. To widen the profile of VPs, an additional veterinary organisation with a younger demographic was invited to distribute the survey link.

The majority of respondents were trained in Ireland, and whilst a small number of VNs and VPs were trained in other countries, it had no significant effect on identification of best outcome in the vignettes. This may reflect a lack of provision in VBM education in other countries, as reported by Kogen et al. [2], Mota-Rojas et al. [3], and Shivley et al. [4].

### 4.2. Confidence with Addressing Cats’ Behavioural Problems

In the timeline of veterinary medicine, advancements in feline medicine, particularly VBM, are relatively recent [6]. As a result, a common narrative in veterinary training was to treat cats as small dogs! The increasing demographic of cat ownership in Europe and elsewhere combined with the relationship between companion animal behavioural problems and animal welfare reflects the importance of veterinary training in feline behaviour. Within this context, one of the first questions in the survey was to ascertain the level of confidence of veterinary professionals in addressing cats’ behavioural problems. Whilst the majority of both VP and VN respondents indicated a reasonable level of confidence (>60%), a sizeable proportion lacked confidence in addressing cats’ behavioural problems. A further analysis, to explore whether confidence influenced the ability to recognise best outcome, showed no effect.

### 4.3. Likelihood of Achieving Best Outcome

Overall, more than 50% of both VPs and VNs selected the response in each vignette that corresponded with our classification of best outcome. Differences between VPs and VNs in recognising best outcomes were observed for three vignettes—spraying (vignette 2), self-mutilation (vignette 4), and resource-based cat aggression (vignette 10). Except for the latter, VP respondents were more likely to correctly select the likelihood of best outcome. It should be noted, however, that although Dettol is not ammonia-based and so would not directly encourage spraying, it would not effectively remove the scent of pre-existing urine marks. In addition, Dettol (a phenolic disinfectant) can damage feline skin or mucous membranes unless allowed to dry completely prior to exposure. This may explain why some respondents labelled this vignette as “Unlikely to achieve best outcome.” Comments from VNs to this vignette included: “Goid [good] advice apart from Dettol, which is phenol-based and toxic to cat”; “Dettol may work after a few applications, however the owner should ensure it is fully dry before allowing the cat back in that area as it is toxic if ingested.” Six comments were recorded for VPs; one mentioned the toxicity of this product to cats.

In addition, the advice provided on self-mutilation could be interpreted as a prescription of treatment. As the prescription of treatment is outside the remit of a VN, they may have found it more difficult to rate the vignette. Level of consensus also varied within veterinary profession; e.g., over 90% VPs recognised that the self-mutilation advice was unlikely to achieve best outcome. Seven VPs made comments on this vignette and four referred to the possibility of it being stress-related; responses included: “There may well be a stress issue here”; “Try and find out if anything is stressing the cat.” Five comments from VNs also referred to stress.

However, only 50% correctly identified the advice in vignette 9 aggression—play-related as being unlikely to achieve best outcome. In contrast, 90.6% of VNs correctly identified that vignette 10 on resource-based inter cat aggression was unlikely to achieve best outcome compared with 50.9% for inappropriate toileting (vignette 1). Nine VNs commented on vignette 10, all recognizing the importance of resources in multicat households; e.g., “Cats don’t like to share resources with other cats.”

### 4.4. Knowledge Gaps

In addition to variation in the level of consensus in correct identification of best outcome, gaps in knowledge were indicated by the selection of the ‘Don’t Know’ and ‘Neither Likely nor Unlikely’ responses. VNs showed a higher level of ‘Don’t Knows’ than VPs, most notably for vignette 4 depicting self-mutilation, although the vast majority of VNs (69%) correctly identified the advice as unlikely to support best outcome. Significantly more VNs selected the Neither Likely nor Unlikely response than VPs for vignettes that did not support best outcome. This may suggest that VNs recognised the flaws and sometimes contradictory advice presented in these vignettes.

The areas with the lowest levels of correct rating of the likelihood of best outcome were for inappropriate toileting (vignette 1), the use of flooding to address nervousness of a kitten towards people (vignette 8) and application of aversive training techniques to treat play-related aggression towards the owner (vignette 9). All of these vignettes were unlikely to support best outcome. In contrast to Shalvey et al. [8] our respondents (53.7%) were less likely to recognise the risks of flooding as an approach to socialisation (cf. Shalvey et al. [8]; 81%). This may be due to greater awareness of socialisation in puppies than kittens. Furthermore, aversive training techniques were depicted in several vignettes in Shalvey et al. [8] and showed that whilst there is good understanding of some forms of positive punishment based training methods such as the use of the check chain in dogs, other types such as the citronella collar were less well understood. Vignette 9 in the current study shows that over 50% of respondents indicated that positive punishment (use of a water pistol) is likely to support best outcome. These findings may indicate that some traditional approaches to behaviour modification persist and have important implications for curriculum design in the area of veterinary behavioural medicine.

### 4.5. Availability of Cat Friendly Practices

According to data collected through the Small Animal Veterinary Surveillance Network (SAVSNET) in the United Kingdom, individual cat owners visit their veterinary clinic 1.2 times less often than dog owners [21]. Concerns about stress may cause cat owners to avoid or delay veterinary visits [22]. This may have serious implications for feline health and welfare. The entire veterinary experience is stressful for cats [23]. However, a number of studies describe interventions to ameliorate this (see [24,25,26,27], for example). In addition, to lessen owner and animal anxiety in veterinary clinics, training and accreditation programmes, such as Fear Free Pets^®^ (founded in 2016) and the International Society for Feline Medicine’s “Cat Friendly Clinics,” have emerged. There are currently four “Cat Friendly Clinics” registered in the Republic of Ireland (https://catfriendlyclinic.org/cat-owners/find-a-clinic/), including two cat-only practices. Veterinary professionals may not be aware of these resources, or may not consider feline stress management to be significant and/or practical [26].

Indeed, the results of our survey indicated that few stress reduction measures are currently in place within respondents’ practices. In the waiting room, the main provisions available are owner distractions (such as magazines and television) and towels/covers for cat carriers. In addition, these are not advertised to clients and this may affect their ability to take advantage of them. Platforms/shelves for cat carriers, separate waiting areas, or cat-only consultation times are not commonly offered. Owner distractions may help to calm clients, and this in turn may reassure their cats. And covering cat carriers may reduce feline stress by preventing visual contact with dogs [27]. However, cats would feel even more secure at a height, above and away from other animals [27]. In addition, cat-only waiting areas or consultation times would be beneficial in preventing most/all sensory contact (visual, auditory, olfactory) with dogs [23]. Veterinary practices simply may not have the space to offer separate canine and feline waiting areas [23,28]. Austrian veterinarians recently rated separate cat waiting areas as highly “important” but poorly “feasible” [29]. In addition, only 36.7% of veterinary practices surveyed in the UK offered separate feline waiting areas [23]. However, with some organisation, it should be possible to run separate consultation times and/or place cat carriers off the ground [25]. Very few respondents indicated that their practice had a cat-only consultation room. Space restrictions and/or patient demographics may preclude this. Only 28.6% of VPs and 39.6% of VNs indicated that they use feline synthetic pheromones in the consultation room. This is perhaps surprising, as they were in use by 77.5% of VPs surveyed in the UK [23]. Cost issues or concerns about efficacy may underlie this. For full efficacy, synthetic pheromones must be used as directed. For example, the Feliway^TM^ diffuser must be switched on constantly and not be blocked by furniture [29]. Our survey also found that cat bags/wraps are not commonly employed during consultations. Anecdotally, some veterinary professionals consider cat bags/wraps to be awkward to use and not very effective. Alternatively, as they are restraint devices, veterinary professionals may feel that it is important to minimise their use [28]. Only 52.4% of VPs and 28.3% of VNs indicated that they allow extra consultation time for nervous cats. The veterinary consultation is typically time pressured. However, a slow and relaxed approach to feline examination generally leads to a more efficient consultation overall [24]. A low stress consultation also yields more valid clinical findings and protects the safety of personnel [24].

Veterinarians in the Arhant et al. [28] study rated a separate cat ward as highly “important” but also highly “unfeasible.” The design of the veterinary establishment likely influences whether complete separation of hospitalised cats and dogs is possible. The majority of our respondents have a cat-only hospital ward at their disposal. In addition, long-term feline patients are kept within the same cage for the duration of their stay and owners are encouraged to bring in familiar items and food from home. This is important, as physical resources and familiar scents are valued by cats and can help them to feel less anxious [30]. However, the majority of our respondents noted that they cannot maintain a set routine within the cat ward, and some indicated that dogs must be walked through the ward for treatment or toileting. In general, this study has identified a low level of integration of feline stress/anxiety reducing options in Irish veterinary practice (particularly in outpatient areas). However, it should be noted that this survey primarily focused on physical/environmental measures, and it is possible that respondents use other methods to reduce feline stress (such as pre-visit training, allowing a cat to exit a carrier themselves, and greeting them calmly; [24]). Our results, coupled with the relative infancy of veterinary education initiatives, such as Fear Free Pets^®^, illustrates that awareness of feline stress reduction must be built and educational programmes further developed and/or promoted.

## 5. Conclusions

Overall, veterinary professionals in Ireland have a good understanding of how to manage feline behavioural problems. However, some confusion remains and confidence in feline behavioural medicine could be improved. Many VPs supported the use of positive punishment and VNs supported an ineffective approach to the management of inappropriate toileting in cats. In addition, “cat friendly” initiatives could be further promoted and developed in Irish veterinary practice. Formal training in VBM as part of the core curriculum, would support an evidence-based approach to recognising common feline behavioural problems, and therapeutic treatments or advice to support the best outcome for feline patients and clients. Day-one competencies in VBM are required to identify best practice and support a consistency in approach, providing veterinary professionals with greater confidence to effectively treat common feline behavioural problems and implement cat friendly initiatives in clinical practice.

## Figures and Tables

**Figure 1 animals-09-01112-f001:**
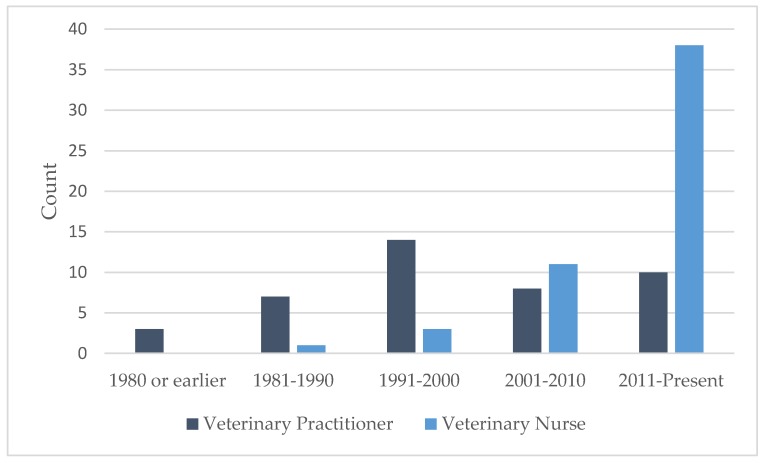
Year of graduation of veterinary practitioner (*n* = 42) and veterinary nurse (*n* = 53) respondents.

**Figure 2 animals-09-01112-f002:**
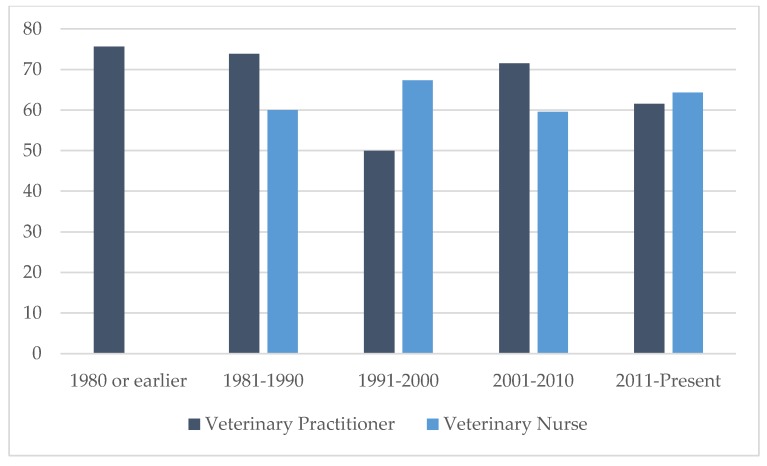
Confidence in advising on feline behavioural issues by profession and decade of graduation.

**Figure 3 animals-09-01112-f003:**
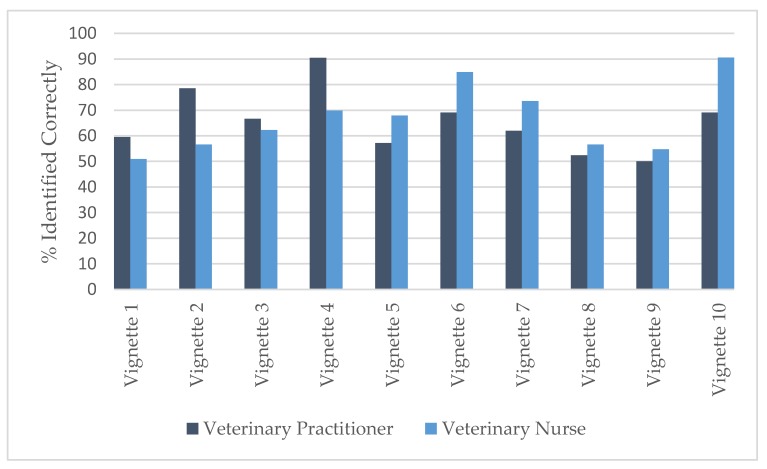
Percentage of veterinary practitioners (*n* = 42) and veterinary nurses (*n* = 53) who identified likelihood of advice to support best outcome for the cat(s) for each vignette. Vignette 1 = inappropriate toileting, 2 = spraying, 3 = destructive behaviour, 4 = self-mutilation, 5 = anxiety—child related, 6 = anxiety—moving home, 7 = fear—loud noises, 8 = fear—strangers, 9 = aggression—play related, 10 = aggression—cat/cat resource-based aggression (see Table 1 for full details).

**Figure 4 animals-09-01112-f004:**
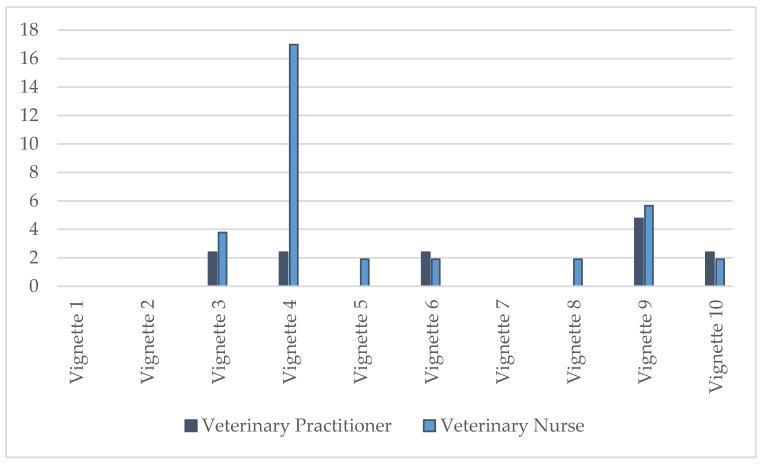
Percentages of veterinary practitioners (*n* = 42) and veterinary nurses (*n* = 53) who answered “Don’t Know” regarding the likelihood of advice in each vignette to support the best outcome for the cat(s) involved. Vignette 1 = inappropriate toileting, 2 = spraying, 3 = destructive behaviour, 4 = self-mutilation, 5 = anxiety—child related, 6 = anxiety—moving home, 7 = fear—loud noises, 8 = fear—strangers, 9 = aggression—play related, 10 = aggression—cat/cat resource-based aggression (see Table 1 for full details).

**Figure 5 animals-09-01112-f005:**
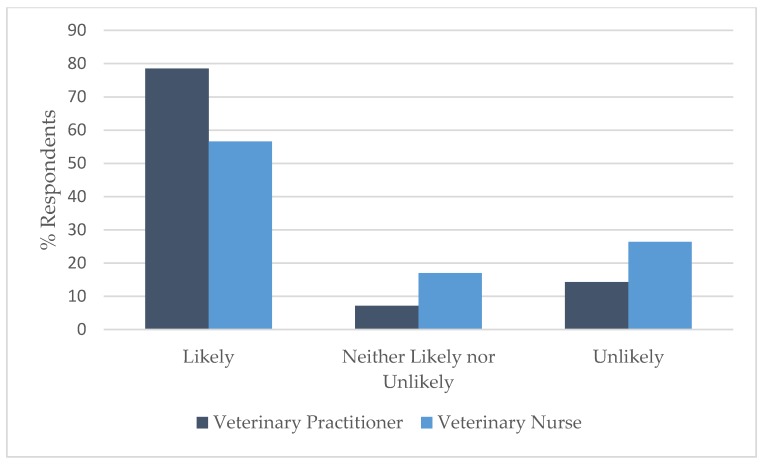
Comparison of the responses of veterinary practitioners (*n* = 42) and veterinary nurses (*n* = 53) regarding the likelihood of advice to support best outcome for the cat(s) involved in Vignette 2: spraying.

**Figure 6 animals-09-01112-f006:**
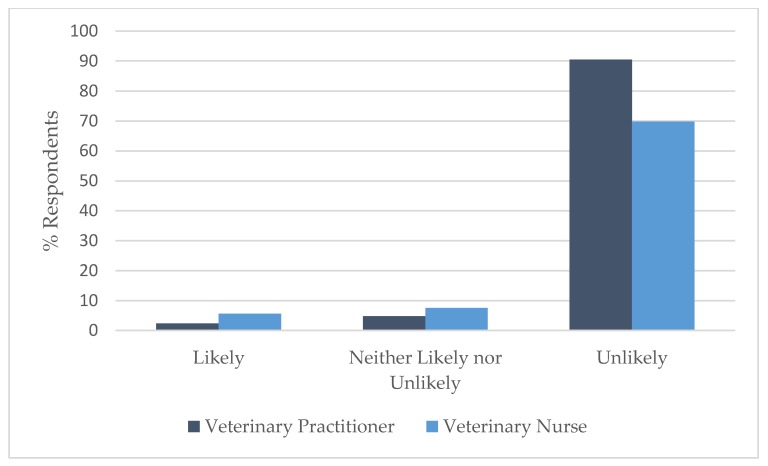
Comparison of the responses of veterinary practitioners (*n* = 42) and veterinary nurses (*n* = 53) regarding the likelihood of advice to support best outcome for the cat(s) involved in Vignette 4: self-mutilation.

**Figure 7 animals-09-01112-f007:**
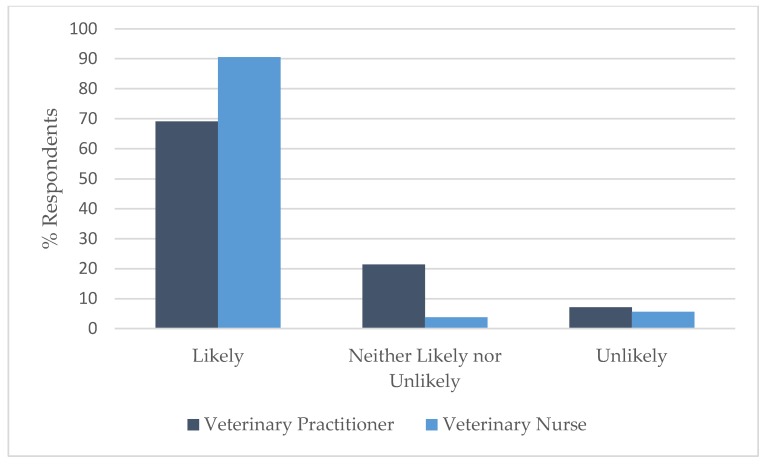
Comparison of the responses of veterinary practitioners (*n* = 42) and veterinary nurses (*n* = 53) regarding the likelihood of advice to support best outcome for the cat(s) involved in Vignette 10: aggression: cat/cat resource-based aggression.

**Table 1 animals-09-01112-t001:** The themes and contents of the ten peer reviewed vignettes depicting advice about common feline behavioural problems from either a veterinary practitioner or a veterinary nurse in each case. The vignettes were designed to be either likely or unlikely to support the best outcome for the cat(s) involved. The best outcome was defined as that which provided a resolution to the behavioural problem while not compromising the animal’s welfare [8].

Theme	Vignette	Likelihood to Achieve Best Outcome
1. Inappropriate Toileting	While at reception after a check-up, Sally asks the vet nurse, Ciara, why her cat has stopped using the litter box saying, “We’ve been having some work done on the house, but he won’t even go when the workers aren’t there.” Ciara tells Sally, “Try moving the litter box to a dark, quiet room away from the work and clean up accidents with any ammonia based cleaner. The harsh smell will encourage him to go elsewhere.”	Unlikely
2. Spraying	John brings in his four year old unneutered cat, Marmalade, because he’s begun spraying next to the back door saying, “I’ve also recently noticed the neighbour’s cat sitting on the garden wall.” The vet advises John to have Marmalade neutered and says, “You should also make sure to clean the spots well with Dettol, spray the area with Feliway and put something up on the garden wall to block the neighbour’s cat.”	Likely
3. Destructive Behaviour	Mary brings in her two year old DSH, Penny, for her yearly check-up. She tells the vet, “Penny won’t quit trying to scratch my new couch instead of her scratching posts. What should I do?” The vet replies, “Spot test and then spray your couch with Feliway to discourage the scratching. You can also use a catnip spray on the scratching posts to encourage Penny to scratch there instead.”	Likely
4. Self-mutilation	Anne brings her elderly cat, Bob, in on a repeat visit for a single, large, crusted, non-healing, self-induced ulcer located between the scapulae. Past examinations have ruled out bacterial, fungal, or parasitic infections as well as other common allergens. Due to the unique presentation of the ulcer and having ruled out most purely medical reasons, the vet diagnoses idiopathic ulcerative dermatitis and tells Anne, “Just keep wrapping the area each time it happens and the ulcer will heal on its own.”	Unlikely
5. Anxiety—Child related	While purchasing flea treatment for her cat, Mary asks the vet nurse, Darren, for advice. “Whenever my nieces come over, my cat spends the whole day avoiding them and will bolt and then vomit up his dinner. What can I do to reduce his anxiety around them? He never settles.” Darren says, “Put up some baby gates, blocking off part of your house from your nieces for the cat. Make sure to feed him in one of these rooms.”	Likely
6. Anxiety—Moving home	During a routine clinical examination, Lorraine asks the vet what she can do to reduce her cat’s anxiety during an upcoming move. The vet suggests, “Get some Feliway diffusers and use them in both houses for at least a few days before the move.”	Likely
7. Fear—Loud Noises	Coming out of a routine consult, Sara asks the vet nurse “Socks always gets so scared of the fireworks. With New Year’s Eve this weekend, is there anything I can do?” The vet nurse tells Sara,“ Make sure to stay in so that you can cuddle and reassure him that everything will be okay.”	Unlikely
8. Fear—Strangers	Clare recently adopted a twelve-week-old kitten, Tommy. After bringing him in for vaccinations and an exam, she asks the vet for advice because Tommy is nervous around guests. The vet says, “The best way to solve this is to introduce Tommy to as many different people as possible so that he gets used to it.”	Unlikely
9. Aggression—Play related	Vanessa has brought her six month old kitten Freckles to the vet for vaccination. She asks how to stop Freckles from attacking her feet. The vet tells Vanessa, “Get a water gun or spray bottle and spray him whenever he jumps on your feet to discourage him.”	Unlikely
10. Aggression—Cat/Cat resource-based aggression	Jo has brought in her two year old cats, Fred and George, for their annual check-up. She asks the vet how to stop Fred from pouncing on and attacking George when he’s done using the litter box and says, “They’ve used the same litter box since they were kittens. It’s only become a problem the last couple of months.” The vet offers her advice, “You need at least two litter boxes for two cats. Try putting in another one, preferably in an area George frequents.”	Likely

**Table 2 animals-09-01112-t002:** Table illustrating the most frequent responses of veterinary practitioners (*n* = 42), veterinary nurses (*n* = 53), and the overall most frequent responses in each vignette. The “Best Outcome” column indicates whether the advice given in a particular vignette would be likely or unlikely to result in best outcome for the cat(s) involved. Higher percentages indicate greater agreement on the likelihood of best outcome depicted in each vignette, while lower percentages represent disagreement. Vignette 1 = inappropriate toileting, 2 = spraying, 3 = destructive behaviour, 4 = self-mutilation, 5 = anxiety—child related, 6 = anxiety—moving home, 7 = fear—loud noises, 8 = fear—strangers, 9 = aggression—play related, 10 = aggression—cat/cat resource-based aggression (see Table 1 for full details).

Vignette	Best Outcome	Most Frequent Response		
		Veterinary Practitioner	Veterinary Nurse	Overall
1. Inappropriate Toileting	Unlikely	59.5% Unlikely	50.9% Unlikely	54.7% Unlikely
2. Spraying	Likely	78.6% Likely	56.6% Likely	66.3% Likely
3. Destructive Behaviour	Likely	66.7% Likely	62.3% Likely	64.2% Likely
4. Self-mutilation	Unlikely	90.5% Unlikely	69.8% Unlikely	77.9% Unlikely
5. Anxiety—Child related	Likely	57.1% Likely	67.9% Likely	62.1% Likely
6. Anxiety—Moving home	Likely	69.0% Likely	84.9% Likely	76.8% Likely
7. Fear—Loud Noises	Unlikely	61.9% Unlikely	73.6% Unlikely	67.4% Unlikely
8. Fear—Strangers	Unlikely	52.4% Unlikely	56.6% Unlikely	53.7% Unlikely
9. Aggression—Play related	Unlikely	50.0% Unlikely	54.7% Unlikely	52.6% Unlikely
10. Aggression—Cat/Cat resource-based aggression	Likely	69.0% Likely	90.6% Likely	81.1% Likely

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
