# Peer review of "Veterinary Professionals’ Understanding of Common Feline Behavioural Problems and the Availability of “Cat Friendly” Practices in Ireland"

_animals, 2019, doi:10.3390/ani9121112_

Round 1

Reviewer 1 Report

y suggestions were followed for the most part.

Reviewer 2 Report

Thank you to the authors for addressing all comments. Nicely done, accept in present form. 

This manuscript is a resubmission of an earlier submission. The following is a list of the peer review reports and author responses from that submission.

Round 1

Reviewer 1 Report

The authors have done a commendable job to describe the status of addressing feline behavior problems in veterinary medicine in Ireland. The design of the study is interesting and innovative and appreciate that the goal for the scenarios is to resolve “the behavioural problem while not compromising the animal’s welfare.” The differences in VP’s and VN’s comfort with feline behavior may reflect differences in education, clinical practice, experience, or more likely, combination of all the factors. My main take-away is that Ireland should provide better education and continuing education on feline behavior and that vet nurses and vet practitioners have different skill sets.

While these are important, there are a number of limitations to this study that make it less of a contribution in the research literature. The development of the survey instrument is interesting, but may not be the best way to assess an individual’s understanding of feline behavior problems. Instead, multiple choice or open ended answers may have been a better way to assess understanding. In addition, the survey should have been “vetted” by a board certified veterinary behaviorist to ensure clarity and consensus on the gold standard answer. Additionally, pooling the answers of extremely likely and likely as well as extremely unlikely and unlikely into categories may be misleading. Statistical analysis should have been done on the answers individually. Furthermore, it would have been nice to know if respondents worked at a feline only practice and if not, what percentage of feline patients are seen in their practice.

Because of these limitations, this manuscript may be better suited as a popular overview article that describes the state of feline behavioral medicine in Ireland rather than a rigorous scientific study. It lacks the correlations and synthesis of parameters that result in compelling implications that contribute to the literature. Regardless, this paper is interesting and authors should be applauded for their efforts to enhance our understanding of veterinary professional’s perspectives on feline behavior.

Abstract

19 – define UCD, spell out veterinarians and veterinary nurses

34 – same

36 – present the exact number instead of “majority”

38 – provide exact p values

Intro

53 – this citation is outdated, but haven’t found any other more recent status updates on behavior:

Juarbe-Díaz, SV (2008).Behavioral medicine opportunities in North American colleges of veterinary medicine: a status report.J Vet Behav.3,4-11

61 – any references to confirm this and citation for it? The number of cat friendly practices or cat only hospitals according to AAFP? – this was referred to in discussion 387-390.

Methods

73 – how was the concept of the survey developed (indicate by whom, initials of authors?)…information from lines 95-99 may be described here. In addition, lines 273-331 may be a better fit in the methods.

77 – 2 individuals should be acknowledged (initials) or described – assume the individuals are specified in Acknowledgements

77 – please include the full survey in supplementary materials (stated in line 113, but not seen in this review)

84 – define out UCD – if this is University of California Davis, what is the relationship between this university and Ireland?

89 – did this have exempt IRB approval?

Table 1 – indicate who determined the gold standard response of “Likelihood to achieve best outcome” – was this determined by board certified veterinary behaviorists?

Scenario 4 is misleading because the medical/derm issue is not fully addressed

120 – what was the purpose of providing these options on the likert scale if they were not going to be examined separately?

Results

131 – This is a significant difference that significance should be calculated – very skewed population. It may be interesting to determine differences in responses between

212 – where was “elsewhere”?

215 – exact numbers of respondents working in cat friendly clinics? – what does this mean? Official certification? Or just practices?

235 – please describe the answers more clearly

Discussion

Lines 273-331 should go in methodology since this was not the result of the survey, but rather the development of the instrument

334 – citation for this? This may be important especially considering that most VN’s graduated after 2010.

351-354 – Please elaborate more why their answers did not correspond to the gold standard answer. It may have been helpful to solicit open ended questions to offer to explain why they chose the answer they did.

441 – what specific day one competencies are recommended from this study?

Author Response

Thank you for your review. Below are the amendments that we have made (also attached).

Abstract

19 – define UCD, spell out veterinarians and veterinary nurses

Amended throughout the manuscript - University College Dublin, veterinary practitioner(s), veterinary nurse(s) where appropriate.

34 – same.

Amended.

36 – present the exact number instead of “majority” –

‘Majority’ was used to provide an overall indication. The actual numbers and percentage varied with vignette and due to the word count restrictions in the abstract we opted for ‘majority’.

38 – provide exact p values.

Amended to p = 0.033, 0.016, and 0.013, respectively.

Intro

53 – this citation is outdated, but haven’t found any other more recent status updates on behavior: Juarbe-Díaz, SV (2008).Behavioral medicine opportunities in North American colleges of veterinary medicine: a status report.J Vet Behav.3,4-11

The two citations were published in the last 3 years:

Mota-Rojas, D.; Orihuela, A.; Strappini-Asteggiano, A.; Cajiao-Pachon, M.N.; AgueraBuendia, E.; Mora-Medina, P.; Ghezzi, M.; Alonso-Spilsbury, M. Teaching animal welfare in veterinary schools in Latin America. Int J Vet Sci Med. 2018, 6, 131-140. DOI: 10.1016/j.ijvsm.2018.07.003.

Shivley, C.B.; Garry, F.B.; Kogan, L.R.; Grandin, T. Survey of animal welfare, animal behavior, and animal ethics courses in the curricula of AVMA council on education-accredited veterinary colleges and schools. J Am Vet Med Assoc. 2016, 248, 1165-1170.

61 – any references to confirm this and citation for it? The number of cat friendly practices or cat only hospitals according to AAFP? – this was referred to in discussion 387-390.

Amended. There is no accreditation programme in Ireland for the registration of cat friendly clinics. Four clinics (increased since the submission of our manuscript in September) are currently registered in Ireland by Cat Friendly Clinic organization (https://catfriendlyclinic.org/cat-owners/find-a-clinic/).

Methods

73 – how was the concept of the survey developed (indicate by whom, initials of authors?)…information from lines 95-99 may be described here. In addition, lines 273-331 may be a better fit in the methods.

Based on your comments we have created a new subsection in the methods about vignette design; as advised we have moved lines 287-390 to this section. Vignettes have been used in a number of other studies at University College Dublin conducted over the last 10 years. The original idea for use of vignettes is from Collins et al 2009 Policy Delphi with vignette methodology as a tool to evaluate the perception of equine welfare.  The Veterinary Journal 2009, 181: 63-69. 

77 – 2 individuals should be acknowledged (initials) or described – assume the individuals are specified in Acknowledgements.

Amended.

77 – please include the full survey in supplementary materials.

The survey has been included as supplementary material.

84 – define out UCD – if this is University of California Davis, what is the relationship between this university and Ireland?

Amended throughout this section

89 – did this have exempt IRB approval?

Amended. The section ethical approval has been moved to the Methods.  

Table 1 – indicate who determined the gold standard response of “Likelihood to achieve best outcome” – was this determined by board certified veterinary behaviorists?

These likelihood to achieve best outcome were based on the scientific literature and were peer reviewed by 2 veterinarians with postgraduate qualifications in animal behaviour. We opted for two veterinary practitioners in Ireland because of the colloquial language (Hiberno-English) used in vignettes. At the time of the study there were no board certified veterinary behaviourists in Ireland.

Scenario 4 is misleading because the medical/derm issue is not fully addressed

We acknowledge the complexity of this issue, the constraints of the vignette design (e.g. word count) meant that scenarios had to be brief. A comment box was provided with each vignette to give respondents the opportunity to qualify their answer.

120 – what was the purpose of providing these options on the likert scale if they were not going to be examined separately?

The Likert Scale was discussed by authors (LG & AH) before the start of the study. We have used a Likert scale to provide respondents with the opportunity to express their level of confidence. Statistical power was the rationale for pooling the likely and unlikely. If the response to the survey had been greater, the Likert scale categories would not have been pooled.

Results

131 – This is a significant difference that significance should be calculated – very skewed population. It may be interesting to determine differences in responses between

This population is consistent with Golden and Hanlon [1] and Shalvey et al [8] and corresponds to when veterinary nursing became offered as a course in Ireland. Due to data protection law, it is challenging to gain access to veterinary practitioners and nurses and the main option is through their representative organisations. Section 4.1 Veterinary Demographic discusses this issue.

212 – where was “elsewhere”?

Elsewhere included those trained in the US, UK, Australia, Germany, Hungary, and Italy. These countries have been added to the text (Section 3.1).

215 – exact numbers of respondents working in cat friendly clinics? – what does this mean? Official certification? Or just practices?

Amended. This refers to practice rather than an official accreditation. The final part of the survey focused on the availability of cat friendly practices in respondent clinics such as the provision of facilities (e.g. shelves and covers for cat carriers) and protocols for the waiting area, consultation room that are likely to alleviate anxiety and stress in cats. Selection of practices was based on recent literature [9, 13].

235 – please describe the answers more clearly

Amended.

Discussion

Lines 273-331 should go in methodology since this was not the result of the survey, but rather the development of the instrument

Amended. A new subsection on vignette design (section 2.2) has been created in the methods.

334 – citation for this? This may be important especially considering that most VN’s graduated after 2010.

We have added a citation Gardiner A 2016 The development and role of the veterinary and other professions in relation to companion animals. In: Sandoe , P.; Corr, S.; Palmer, C. (Editors) Companion Animal Ethics, 1st ed.; Wiley-Blackwell.

351-354 – Please elaborate more why their answers did not correspond to the gold standard answer. It may have been helpful to solicit open ended questions to offer to explain why they chose the answer they did.

A comment box was provided with each vignette, to give respondents the opportunity to qualify their answers. A selection of respondent comments have now being included in the discussion to reflect respondent understanding of the issues.

441 – what specific day one competencies are recommended from this study?

It was beyond the scope of this study to recommend specific day one competences in VBM. However the results are part of a portfolio of evidence indicating that there is a need to develop such competences to support curriculum design. In the context of the current study this would provide veterinary professionals with greater confidence to effectively treat common feline behaviour problems and implement cat friendly initiatives in clinical practice. The conclusions have been amended to reflect the relevance of current findings to the development of Day One Competences.

Reviewer 2 Report

 This is a study, based on a survey, of how vets and vet nurses treat common feline behavior problems. Vet is too informal for a scientific journal and is confused with veterans

 In general this is a well-written paper and although the results are discouraging it should lead to more teaching of behavior at veterinary colleges and at veterinary conferences.

 UCD is probably not University of California Davis. For this international audience use full name the first time UCD is mentioned

 PDSA define

Feline is used for cat and canine for dog; feline and canine are adjectives an adjective

  108 Spraying   Dettol? Aren’t there specific remedies for cleaning up urine available in Ireland?   Urine Off, Anti Icky Poo Nature’s Miracle etc.

  Most of the studies on efficacy of Feliway are flawed. Perhaps some of your respondents knew that and therefore, did not see that treatment with it was likely to help.Frank, D., G. Beauchamp & C. Palestrini. 2010. Systematic review of the use of pheromones for treatment of undesirable behavior in cats and dogs. Sci Rep. 236(12): 1308-1316.

.

 Destructive behavior There is a new product that encourages scratching called Feliscratch

Over grooming. An area between the scapulae is unlikely to be reached by the cat.

424 interestingly

Author Response

Thank you for your review. Below are the amendments that we have made to address your recommendations.

This is a study, based on a survey, of how vets and vet nurses treat common feline behavior problems. Vet is too informal for a scientific journal and is confused with veterans.

Amended.

 In general this is a well-written paper and although the results are discouraging it should lead to more teaching of behavior at veterinary colleges and at veterinary conferences.

 UCD is probably not University of California Davis. For this international audience use full name the first time UCD is mentioned.

Amended.

PDSA define

Amended.

Feline is used for cat and canine for dog; feline and canine are adjectives an adjective

  108 Spraying   Dettol? Aren’t there specific remedies for cleaning up urine available in Ireland?   Urine Off, Anti Icky Poo Nature’s Miracle etc.

Dettol is a common non-ammonia based disinfectant available in Ireland that an owner would be likely to have at home. It is however phenol-based and potentially toxic to cats. We have included respondent comments, which reflect this issue.

  Most of the studies on efficacy of Feliway are flawed. Perhaps some of your respondents knew that and therefore, did not see that treatment with it was likely to help. Frank, D., G. Beauchamp & C. Palestrini. 2010. Systematic review of the use of pheromones for treatment of undesirable behavior in cats and dogs. Sci Rep. 236(12): 1308-1316.

We acknowledge that additional, more rigorous studies are needed to properly assess the effectiveness of Feliway. However, following a meta-analysis, Mills et al. (2011) concluded that Feliway is effective at reducing the incidence of urine sprayinga. In addition, the International Society for Feline Medicine (ISFM) recommends using Feliway to support the treatment of urine sprayingand to reduce environmental or social stressc . Comment boxes were provided after each vignette but our respondents did not note any concerns about Feliway’s effectiveness.

aMills, D.S., Redgate, S.E. & Landsberg, G.M. (2011) A meta-analysis of studies of treatments for feline urine spraying.’ Plos One 6(4), e18448

bCarney, H.C., Sadek, T.P., Curtis, T.M., Halls, V., Heath, S., Hutchison, P., Mundschenk,  K. and Westropp, J.L. (2014) AAFP and ISFM guidelines for diagnosing and solving house- soiling behavior in cats. Journal of Feline Medicine and Surgery 16(7), 579-598.

cEllis, S.L.H., Rodan, I., Carney, H.C., Heath, S., Rochlitz, I., Shearburn, L.D., Sundahl, L.D. and Westropp, J.L., (2013) AAFP and ISFM feline environmental needs guidelines. Journal of Feline Medicine and Surgery 15(3), 219-230.

Destructive behavior There is a new product that encourages scratching called Feliscratch.

Feliscratch was launched in Ireland in 2017. It is not mentioned in vignette as we thought respondents would not be familiar with it.

Over grooming. An area between the scapulae is unlikely to be reached by the cat.

We have renamed the vignette to self-mutilation because the aim of this scenario was to describe a stress or anxiety-related behaviour. Per Titeux et al [17] this is one of three locations that are characteristic of idiopathic ulcerative dermatitis lesions.

424 interestingly

Deleted – interestingly.

Reviewer 3 Report

Observations of veterinary professionals’ understanding of treating common feline behaviour problems and the availability of “cat friendly” practices in Ireland

The paper attempts to evaluate the extent of an important knowledge gap (in re: ability to appropriately interpret feline behavior, offer pertinent treatments, and awareness of treatment-related resources) in the veterinary profession in Ireland. This is a worthy contribution. Given the broad reach of the journal, the paper should make an attempt to contextualize the need for such an evaluation and its results in a more globally-relevant approach. Is this a systemic problem across similar cat-keeping societies? What can be said about the consistency, relevance, and quality of veterinary training? Etc.

As addressed more specifically in the comments below, the paper would benefit from reorganization of information. 

**Where is the information about ethics approval in the paper?

Title:

Line 2. A bit misleading. ‘Observations’ can be interpreted a number of ways. Survey should be mentioned in title. Perhaps a shorter title?

Simple summary

Line 10-11. Minor point. Opening sentence of simple summary is a bit clumsy.

Line 19: Minor point. Typically, unconventional to begin a sentence with a numeral (also see line 42).

Abstract

Lines 25-30. Writing is a bit unclear, e.g., the writing should be more precise than ‘come into focus’.

Line 39: I know Abstracts have strict word limits, however the discussion/interpretation aspect of Abstract is quite thin.

Introduction

Line 51. Minor points. A) Need period after et al. (see other instances). B) If speaking of a proportion (e.g., half) of sample group, then ‘less’ should be used. If speaking of the number within a sample group then ‘fewer’ should be used.

Lines 52, 54, 55. Minor point. Consistency of spelling (i.e., program/programme).

Lines 56-59. Authors should re-read to ensure the sentences are effectively conveying the intended idea. The idea that cats weren’t given their proper due in vet medicine is implied rather than clearly stated in this section. Perhaps better to open up paragraph directly speaking about the (evolving) status of cats in vet medicine.

Line 64. Minor. Should be fear-related.

Overall. The introduction would benefit from greater inclusion of the types of behaviors (beyond fear) that vets and people may find problematic and symptomatic of other issues.

Line 64. “…understanding of advice to support…” Is this really the aim of the study?  This doesn’t make sense. Cf. opening sentence of Methods section.

Methods

Aside from veterinary professions, was a social scientist, expert in survey design, consulted?

Table 1. The formatting of Table 1 could be more streamlined.

Table 1. Is there a reason why vet nurses are given identities in the vignettes, but vets remain anonymous?

Line 91. Capitalisation of ‘o’ in other needed to correspond with ‘Other’ category.

Line 92. Was there a way to determine if confidence in addressing feline behavior was accurately assessed?

Line 71/112. Given the deference given to Fear Free Initiative, perhaps greater elaboration is needed in the paper to contextualize the survey.

Line 118. The exclusion of “Don’t know” data seems problematic. Because Wilcoxon requires ordinal data is not reason to exclude, but perhaps a sign that this may not be the most appropriate test for the data set.

Results

Line 130. No apostrophe in 1990s.

Line 138/Line 142. This is too general of a description. Better to actually describe the data and cite the figure being referred to parenthetically.

Where there were no significant differences on comparison, might a table be a more streamlined presentation of this information?

Line 223/Section 3.7. Much of this is methodological. This section should be reviewed to move appropriate information to the Methods section.

Discussion

Line 257, etc. This section seems more appropriate for Methods. Perhaps some of the information should even be presented in the Introduction.

Lines 268-269. What is ‘recently’ referring to? Invited for this paper? Invited after this survey was distributed?

Line 271. ‘no impact on results’ is too vague. The reader can infer the meaning, but better to be explicit.

Section 4.2. Again, there is information - re: rationale for vignette design - here that is more appropriate for the Methods. Perhaps there should be a section in the Methods dedicated to this.

Line 274. Briefly remind the reader what type of surveys the vignettes are derived from.

Can the paper provide recommendations for improvement of knowledge, knowledge dissemination?

Author Response

Thank you for your review. Below are the amendments that we have made to address your recommendations.

**Where is the information about ethics approval in the paper?

The ethical approval reference has been moved to the methods section.

Title:

Line 2. A bit misleading. ‘Observations’ can be interpreted a number of ways. Survey should be mentioned in title. Perhaps a shorter title?

The title has been shortened (and ‘observations’ deleted)

Simple summary

Line 10-11. Minor point. Opening sentence of simple summary is a bit clumsy.

Added some punctuation to sentence to enhance readability.

Line 19: Minor point. Typically, unconventional to begin a sentence with a numeral (also see line 42).

Amended.

Abstract

Lines 25-30. Writing is a bit unclear, e.g., the writing should be more precise than ‘come into focus’.

Amended.

Line 39: I know Abstracts have strict word limits, however the discussion/interpretation aspect of Abstract is quite thin.

The word count restrictions mean that there is limited scope to include additional information. The final sentence has been amended.

Introduction

Line 51. Minor points. A) Need period after et al.

Amended 15 instances (see other instances).

B) If speaking of a proportion (e.g., half) of sample group, then ‘less’ should be used. If speaking of the number within a sample group then ‘fewer’ should be used.

Amended.

Lines 52, 54, 55. Minor point. Consistency of spelling (i.e., program/programme).

Amended throughout.

Lines 56-59. Authors should re-read to ensure the sentences are effectively conveying the intended idea. The idea that cats weren’t given their proper due in vet medicine is implied rather than clearly stated in this section. Perhaps better to open up paragraph directly speaking about the (evolving) status of cats in vet medicine.

Amended (lines 80-81 and 93-95).

Line 64. Minor. Should be fear-related.

Amended

Overall. The introduction would benefit from greater inclusion of the types of behaviors (beyond fear) that vets and people may find problematic and symptomatic of other issues.

Amended and new reference cited.

Line 64. “…understanding of advice to support…” Is this really the aim of the study?  This doesn’t make sense. Cf. opening sentence of Methods section.

The methods have been amended.

Methods

Aside from veterinary professions, was a social scientist, expert in survey design, consulted?

A social scientist was involved in an earlier study, which developed the use of vignettes to explore stakeholder perception of equine welfare issues in Ireland:

Collins, J.A., Hanlon, A., More, S.J., Wall, P.G., Duggan, V. Policy Delphi with vignette methodology as a tool to evaluate the perception of equine welfare. The Veterinary Journal 2009, 181, 63-69. DOI:10.1016/j.tvjl.2009.03.012

Vignette methodology has subsequently been used in two further studies – on veterinary ethics and common dog behaviour problems.

Magalhaes-Sant’Ana, M., Hanlon A.J., Straight from the Horse’s Mouth: Using vignettes to support student learning in veterinary ethics. J Vet Med Educ. 2016 DOI: 10.3138/jvme.0815-137R

Shalvey, E.; McCorry, M.; Hanlon, A. Exploring the understanding of best practice approaches to common dog behaviour problems by veterinary professionals in Ireland. Ir Vet J. 2019. DOI: 10.1186/s13620-019-0139-3.

Table 1. The formatting of Table 1 could be more streamlined.

The table has been formatted by journal (and differs to our original submission).

Table 1. Is there a reason why vet nurses are given identities in the vignettes, but vets remain anonymous?

Only one of the three vignettes depicting a veterinary nurse did not give the nurse an identity. This was unintentional and may have been due to trying to keep the vignette short.

Line 91. Capitalisation of ‘o’ in other needed to correspond with ‘Other’ category.

Amended.

Line 92. Was there a way to determine if confidence in addressing feline behavior was accurately assessed?

With section 3.2, we attempted to determine if those who had rated themselves to be confident were more likely to agree with our determinations for the vignettes than those who had rated themselves lower on the scale (>75 & <50) but we didn’t find any significant results.

Line 71/112. Given the deference given to Fear Free Initiative, perhaps greater elaboration is needed in the paper to contextualize the survey.

Reference to the Fear Free Initiative was to provide an example only. Based on your comment we have amended references to Fear Free.

Line 118. The exclusion of “Don’t know” data seems problematic. Because Wilcoxon requires ordinal data is not reason to exclude, but perhaps a sign that this may not be the most appropriate test for the data set.

Our understanding is that it is not unusual to handle ‘don’t knows’ as missing values and exclude from data analysis. It is debatable as discussed by Iannario et al 2018 Ordinal data models for no-opinion responses in attitude survey. Sociological Methods & Research 1-27 doi:10.1177/0049124118769081

Results

Line 130. No apostrophe in 1990s.

Amended.

Line 138/Line 142. This is too general of a description. Better to actually describe the data and cite the figure being referred to parenthetically.

A more detailed description has been provided for the level of confidence with addressing cat behaviour problems.

Where there were no significant differences on comparison, might a table be a more streamlined presentation of this information?

Sections 3.4, 3.5 and 3.6 have been merged (and edited) with other relevant sections in the results to streamline the results section.

Line 223/Section 3.7. Much of this is methodological. This section should be reviewed to move appropriate information to the Methods section.

Amended.

Discussion

Line 257, etc. This section seems more appropriate for Methods. Perhaps some of the information should even be presented in the Introduction.

We were not sure if this comment referred to Line 261-262 or line 277 (section 4.2 vignette selection). Lines 261-262 have been moved to the introduction. Based on the comments of reviewer 2 we have moved Vignette Selection to the methods.  

Lines 268-269. What is ‘recently’ referring to? Invited for this paper? Invited after this survey was distributed?

Amended.

Line 271. ‘no impact on results’ is too vague. The reader can infer the meaning, but better to be explicit.

Amended.

Section 4.2. Again, there is information - re: rationale for vignette design - here that is more appropriate for the Methods. Perhaps there should be a section in the Methods dedicated to this.

Amended. We have created a new subsection in the methods on vignette design.

Line 274. Briefly remind the reader what type of surveys the vignettes are derived from.

Can the paper provide recommendations for improvement of knowledge, knowledge dissemination?

We believe that the findings will be helpful to veterinary educators when designing or modifying teaching on feline behaviour. Indeed, the authors have already adapted their own teaching based on the results of the two previous surveys (Golden & Hanlon 2018 and Shalvey et al 2019). Our findings also indicate that continuous professional development (CPD) offerings are needed in the areas of feline behavioural problems and cat friendly practices. The vehicles for CPD offerings could include conferences, seminars, and online education provided by universities or veterinary behaviour organisations/interest groups.

Golden, O.; Hanlon, A.J. Towards the development of day one competences in veterinary behaviour medicine: survey of veterinary professionals experience in companion animals practice in Ireland. Ir Vet J.2018, 71, 12. DOI 10.1186/s13620-018-0123-3.

Shalvey, E.; McCorry, M.; Hanlon, A. Exploring the understanding of best practice approaches to common dog behaviour problems by veterinary professionals in Ireland. Ir Vet J. 2019. DOI: 10.1186/s13620-019-0139-3.
